# Design and Validation of a Scale Measuring Attitudes toward Refugee Children

**Georgia Angelidou *** , **Eva María Aguaded-Ramírez and Clemente Rodríguez-Sabiote**

Department of Research Methods and Diagnostics in Education, University of Granada, 18071 Granada, Spain; eaguaded@ugr.es (E.M.A.-R.); clerosa@ugr.es (C.R.-S.)
* Correspondence: g_aggelidou@hotmail.com; Tel.: +30-6973-510-488

**Abstract:** The aim of this study was to design, develop, and validate a questionnaire evaluating attitudes toward refugee children. Method: The questionnaire was analyzed using SPSS Software version 21.0 (IBM Corp., 2012, Armonk, NY, USA). Results: Cronbach's $\alpha$ was greater than 0.9. According to an expert's review, the instrument provided arithmetic means higher than 2.5, the dimensions evaluated had interclass correlation coefficients greater than 55, and the corrected correlation values of the item-total were greater than 0.35. Conclusions: The Attitude Questionnaire toward Refugee Children was found to be an adequate instrument for better understanding and measuring the attitudes of host countries' citizens toward refugee children.

**Keywords:** scale validation; refugee children; education

## 1. Introduction

Since 2014, the total number of global displaced people has increased, meaning many people are in need of protection and asylum around the world [1]. The annual Global Trends study published by the United Nations High Commissioner for Refugees (UNHCR) highlighted that, in 2017, as a result of conflict, persecution, and generalized violence, 68.5 million people around the world were forcibly displaced, of which 24.5 million were refugees, 40 million were internally displaced persons, and 3.1 million were asylum-seekers [2].

Many countries have shown interest in hosting refugees. For instance, in 2017, the USA received 815,600 migrants and refugees, while Greece received 94,600 and Spain received 28,200. Among adults, there are thousands of children seeking international protection. Specifically, according to various United Nations agencies, it is estimated that half of the world's refugee population is made up of children [3–6].

Apart from receiving and hosting refugees, host countries are facing a new challenge: helping these people integrate into host societies and provide them education and development opportunities. According to the 2030 Agenda for Sustainable Development [7], among the international migrants and refugees, there are 31 million school-aged children whose education is considered a long-term investment and strategic priority, as education plays an important role in economic mobility, learning outcomes and generally, in social integration [7].

Regardless of government policy, it is important for the foreign population to be accepted by the host society. Currently, there is a shortage of studies about the attitudes of citizens in host countries toward refugee children. For this reason, we tried to fil this gap and aimed to design, develop, and validate a questionnaire to measure the attitudes of adults in host countries toward refugee children.

## 2. Literature Review Summary

### 2.1. Resettlement Schemes

Due to the war in Syria, the armed conflicts both in Afghanistan and the Middle East, and the strengthening of ISIS (Islamic State of Iraq and the Levant), Europe experienced its greatest refugee crisis since World War II. In 2016 alone, approximately one million people attempted to land in Europe, seeking asylum and international protection. Most of them were from Syria, Iraq, and Afghanistan, who reached Europe by crossing the Mediterranean Sea. Since then, the increase in the number of refugees in Europe has been significant. For instance, during the first three months of 2018, Greece received 4214 asylum seekers and Spain received 2932 [8].

Under such circumstances, in September 2015, a relocation scheme was introduced by Council Decisions (EU) 2015/1523 and 2015/1601 for around 160,000 asylum seekers. The relocation program was designed as an emergency measure to alleviate pressure on Greece and Italy, constituting a partial derogation to the Dublin Regulation. According to the Asylum Information Database (2018):

Out of the target of 66,400 asylum seekers to be relocated from Greece, 21,731 had effectively been transferred as of 28 January 2018. The European Commission has been regularly reporting on the scheme, highlighting a number of challenges resulting in slow and inefficient implementation of Member States' commitments [9].

Since September 2015, 30,836 places for relocation were offered to the Greek State by the following countries: Belgium, Bulgaria, Croatia, Cyprus, Czech Republic, Estonia, Finland, France, Germany, Island, Latvia, Lichtenstein, Lithuania, Luxembourg, Malta, Netherlands, Norway, Portugal, Romania, Slovakia, Slovenia, Spain, and Switzerland [9].

The United States has historically played an important role in refugee resettlement, and remains one of the most important countries in terms of relocation and resettlement issues. In the 2016 fiscal year, the United States resettled 84,994 refugees, providing them international protection [10]. Apart from the USA, Puerto Rico, which is unincorporated territory of the United States, was not excluded from the Refugee Resettlement Programme. In 2015 the State Department of the Government of Puerto Rico agreed to join the Programme, which would become part of next fiscal year (2016) [11].

### 2.2. Integration: A Two-Way Process

The integration of refugees is a two-way process. An inclusive and tolerant society is a prerequisite for the successful integration of refugees. Firstly, the acceptance of refugees by the host countries' citizens and the existence of a general sense of being welcome play an important role in integrating refugees and helping them start a new life in the community. When host communities allow refugees to rebuild their lives, the refugees can enrich the society. Promoting a tolerant society is also linked to the fight against discrimination and racism. Secondly, refugees should be willing to be integrated into the host society, access services, educate themselves, learn the language of the host country, and overcoming new challenges [12].

The integration and inclusion process can be a difficult process as the integration in the host society means the refugees must abandon their former life and start a new one. There may be considerable cultural differences between locals and refugees and various problems can arise, including unemployment, difficulty in accessing education, racism, negative attitudes, etc. [12].

According to the 1951 United Nations Convention Relating to the Status of Refugees and the 1967 Protocol, refugees, both adults and minors, have a series of fundamental rights in host countries. One of these rights is the right to education. Education is not only important for the better integration of individuals but can also benefit the entire society [13,14]. However, the host communities may not be ready to accept refugees and respect their rights and the host parents may not be comfortable with accepting refugee children in educational centers. As professionals and researchers who are dedicated to the field of education and migration, we considered it important to examine the attitudes of host

countries toward refugees and especially toward refugee children to answer the above questions and better understand how locals face forced migration.

*2.3. Literature Review*

We considered it important to define the concept of attitude. Attitude is the tendency toward a social object, derived by the feelings, thoughts, and behaviors toward it. Attitude can either be a positive or negative predisposition toward objects or people, which is why humans often hold attitudes toward a variety of objects or symbols, such as attitudes toward family, a teacher, abortion, economic policy, work, etc. Attitudes are related to the behavior we maintain toward people or objects. Attitudes are only an indicator of a given behavior, but not the behavior itself. Therefore, attitude measurements should be interpreted as symptoms and not as facts [15].

We must remember that attitudes have two distinctive features: (1) the direction (positive or negative) and (2) the intensity (high or low). These features should be part of the attitude measurement process. "Attitudes are not susceptible to direct observation but must be inferred from verbal expressions or observed behavior" [16]. "This indirect measurement is carried out using some scales starting from a series of affirmations, propositions, or judgments on which individuals express their opinion; based on this information, we deduce or infer attitudes" [17].

Attitudes are measured through instruments. Scales are instruments designed to measure the properties of individuals or groups, and they are widely used to measure attitudes and values. These scales have to: (1) be used as an instrument to measure the characteristics of a variable and scales allow the presentation of the values of the variable by a score, (2) be used as an operational definition of an abstract concept and (3) be used as an instrument for measuring complex or sensitive issues [18].

The development of a measuring instrument is a difficult task. Its design and validation, as an instrument aimed to evaluate human attitudes, is complex and should not be chaotic, as long as specific rules are followed based on sufficient data organization. One of the most common errors during the development of an attitude scale instrument is the development of a bank of items which has not undergone any test of scientific rigor. Effectively, the final instrument is not validated, thus affecting reliability and the ability to generalize the research results [18].

According to a literature review, several authors have researched host country attitudes toward foreign populations and their research contributions have been significant. Specifically, several researchers developed some attitude scales for immigrant children [19,20]. In their studies, they question the population's sensitivity to immigrants [21], or host countries' attitudes toward immigrants [22,23]. However, no attitude scales for refugee children have been created to date.

As human beings, we have our own beliefs, values, and attitudes, and these could affect our behavior toward others. As teachers often work with pupils and students from different countries and cultures, the host country's lifestyle may different from the children's own or even appear unacceptable. In our case, the target group was refugee children, whose number has increased worldwide since 2015. If we want to help them integrate and provide them services that meet their expectations and needs, we need to be aware of our own attitudes, because integration is a two-way process. In other words, if we want to create integration plans, it is essential to work with both sides, and to first of all to determine locals' attitudes. Based on the mentioned reasons, we considered the need to design and validate a scale to measure the attitudes toward refugee children.

Thus, the primary objective of this article is to present the validation analysis of a scale designed to assess attitudes toward refugee minors. This objective is achieved through: (1) the design of a basic scale of assessment of host countries' attitudes toward refugee children and (2) validation of the scale.

With this validation, we attempted to create an instrument that is valid, reliable, functional, effective, and understandable. The validation of data collection instruments, their importance, and necessity has been the object of study of many investigations [24–26].

## 3. Materials and Methods

### 3.1. Design

A design validation test was used [27–29] for the validation of the Attitudes Questionnaire toward Refugee Children (AQReC), which was created ad hoc. It consists of the calculation of the quality parameters contemplated by the Classical Test Theory (CTT) and the Generalizability Theory (GT), by its application to a pilot sample ($N = 535$) and valued by a group of experts ($N = 6$).

### 3.2. Sampling

The scope of the current study was international. The pilot sample was selected with a focus on participants from different countries, cultures, and religions so it could be applied to different cultural contexts. More specifically, we chose Greece, Spain, and the USA, which have traditionally been some of the most important countries in terms of emigration. We faced many difficulties in accessing USA citizens and, for this reason, we conducted the research in Puerto Rico, as it is an unincorporated territory of the United States, and the access for our scientific team was easier. As a result, the population consists of Spanish, Puerto Rican, and Greek adults of both sexes.

We used convenience sampling, a non-probability technique, where the subjects are chosen because of the proximity to the researcher and the convenient accessibility [27]. The sample consisted of 180 Spanish, 140 Puerto Ricans, and 215 Greeks, both women and men, over 18 years old; however, we focused our study on university students. The pilot sample size used in the validation of the questionnaire mentioned above (AQReC) was 535 participants, which is more than enough for this type of study (pilot study), as pilot samples usually do not exceed 100 participants [28]. The sample's profile is detailed in Table 1.

**Table 1.** Sample profile.

| Sample Profile | |
| --- | --- |
| **Sex** | |
| Male | 30.6% |
| Female | 68.4% |
| Prefer not to reveal: | 1% |
| **Age** | |
| 18–24 | 26.6% |
| 25–35 | 53.9% |
| 36–55 | 16.2% |
| >55 | 3.3% |
| **Nationality** | |
| Spanish | 32.7% |
| Greek | 41.1% |
| Puerto Rican | 26.2% |
| **Religion** | |
| Greek Orthodox | 41.8% |
| Catholic | 19.5% |
| Agnostic | 11.1% |
| Atheist | 21.2% |
| Muslim | 0.5% |
| Other | 5.8% |
| **Educational Level** | |
| Primary Education | 0.2% |
| Secondary Education | 8.4% |
| University Studies | 43.8% |
| Master's Studies | 39% |
| PhD Studies | 7.6% |
| Other studies | 1% |

### 3.3. Instrument Validation

The validated questionnaire consisted of 34 items, organized in two subscales. The first subscale consisted of 24 items, including items about locals' attitudes toward refugee children. These items were designed using the Likert scale, with five response options: strongly disagree, disagree, neutral, agree, and strongly agree. The second subscale consisted of 10 items measuring feelings toward refugee minors, also with five response options on a Likert scale: very week, weak, none, strong and very strong.

### 3.4. Procedure

The first part of the procedure was related to processing the data and the validation of the questionnaire (analysis of reliability-internal consistency and validity of concurrent criteria). Subsequently, the questionnaire was submitted for evaluation to six experts (professors) to guarantee the validity of the content of the items in terms of content, writing, understanding, and relevance.

## 4. Results

All reliability and validity analyses were performed with SPSS Software version 21.0 (IBM Corp., 2012, Armonk, NY, USA).

### 4.1. Reliability Results as Internal Consistency

As the instrument, used for the data collection (questionnaire), was only administered the participants once, the type of reliability cannot be other than the assurance of internal consistency. The most appropriate coefficient for calculation and assessment is Cronbach's $\alpha$. The values obtained for each item are listed in Table 2.

**Table 2.** Cronbach's $\alpha$ values.

| Dimension | No. of Items | Cronbach's $\alpha$ | Cronbach's $\alpha$ Based on Standardized Items |
|---|---|---|---|
| **Attitude Toward Refugee Minors** | **24** | **0.919** | **0.924** |
| Refugee minors can spread diseases to schools in my country. | | Cronbach's $\alpha$ if item deleted 0.921 | |
| If I adopted a child, it would bother me if the child was from another country. | | Cronbach's $\alpha$ if item deleted 0.918 | |
| It would be a problem if my son/daughter had a refugee child as a classmate. | | Cronbach's $\alpha$ if item deleted 0.916 | |
| Refugee children cause troubles during class. | | Cronbach's $\alpha$ if item deleted 0.917 | |
| It would be a problem if my son/daughter had a refugee child as a friend. | | Cronbach's $\alpha$ if item deleted 0.916 | |
| All people should be respected the same. | | Cronbach's $\alpha$ if item deleted 0.918 | |
| Refugee minors have the right to talk about their religion. | | Cronbach's $\alpha$ if item deleted 0.915 | |
| Refugee children have the right to celebrate their religious holidays in my country. | | Cronbach's $\alpha$ if item deleted 0.914 | |
| The refugee minors, in the future, will want to impose their religion on us. | | Cronbach's $\alpha$ if item deleted 0.917 | |
| We have to respect people from other countries. | | Cronbach's $\alpha$ if item deleted 0.916 | |
| The children of the refugees are inferior to the children of the Spaniards. | | Cronbach's $\alpha$ if item deleted 0.918 | |

**Table 2.** *Cont.*

| | |
|---|---|
| Refugee children have the right to use their native language. | Cronbach's α if item deleted<br>0.915 |
| Our country should expel all refugee children. | Cronbach's α if item deleted<br>0.916 |
| Due to refugee children, we can learn about other cultures and languages. | Cronbach's α if item deleted<br>0.913 |
| Thanks to refugee children, we can learn new things. | Cronbach's α if item deleted<br>0.912 |
| All refugee children who come to my country are poor. | Cronbach's α if item deleted<br>0.923 |
| Thanks to refugee children, we can learn new customs. | Cronbach's α if item deleted<br>0.913 |
| We must behave toward refugee children in the same way we do toward Spaniard children. | Cronbach's α if item deleted<br>0.915 |
| Refugee children will cause an increase on crime rate in my country. | Cronbach's α if item deleted<br>0.916 |
| Refugee children arrive in my country only for economic reasons. | Cronbach's α if item deleted<br>0.920 |
| Refugee children have the right to attend school classes along with Spaniard children. | Cronbach's α if item deleted<br>0.913 |
| Public services should focus more on welcoming refugee minors. | Cronbach's α if item deleted<br>0.917 |
| Schools should have educational programs to meet the needs of refugee minors. | Cronbach's α if item deleted<br>0.913 |
| All children have the right to education. | Cronbach's α if item deleted<br>0.916 |

| Dimension | No. of Items | Cronbach´s α | Cronbach's α Based on Standardized Items |
|---|---|---|---|
| **Feelings Toward Refugee Minors** | **10** | **0.678** | **0.680** |
| I feel admiration toward refugee minors. | | Cronbach's α if item deleted<br>0.620 | |
| I feel hate toward refugee minors. | | Cronbach's α if item deleted<br>0.673 | |
| I feel compassion toward refugee minors. | | Cronbach's α if item deleted<br>0.639 | |
| I feel sorry toward refugee minors. | | Cronbach's α if item deleted<br>0.673 | |
| I fear refugee minors. | | Cronbach's α if item deleted<br>0.658 | |
| I feel solidarity with refugee minors. | | Cronbach's α if item deleted<br>0.640 | |
| I feel indifferent toward refugee minors. | | Cronbach's α if item deleted<br>0.653 | |
| I am curious about refugee minors. | | Cronbach's α if item deleted<br>0.668 | |
| I feel concern toward refugee minors. | | Cronbach's α if item deleted<br>0.618 | |
| I dislike refugee minors. | | Cronbach's α if item deleted<br>0.689 | |

According to Table 2, the Cronbach's α values obtained in each dimension are different. Thus, in the attitude dimension, the value of Cronbach's α increased to 0.914, slightly improved, when considering the standardized item scores (Cronbach's α based on standardized items) with a value of 0.924. The sensitivity dimension obtained a Cronbach's α of 0.678, similarly slightly improved, when we consider Cronbach's α based on standardized items of 0.680. In the first case, as a general criterion, the Cronbach's α values, both regular and standardized, can be considered excellent and indicate the high internal consistency of the test [29]. In the second case, according to the sensitivity dimension, we obtained values, both normal and standardized, close to 0.7. The values are significantly lower but minimally adequate and acceptable [30–32]. The values of Cronbach's α (if item deleted) show that the value of the test's reliability would not improve, which is why its presence on the scale is justified.

*4.2. Results for Content Validity*

To validate the content of the questionnaire, we used expert judgment. The objective was a two-scale strategy. The first scale involved the determination of the degree of convenience of the items (34 contemplated items), and the second scale involved the degree of judgment agreement issued by the experts, although it is known that both are closely related [33,34].

First, we highlighted the degree of convenience of the items measured on a scale of 1 to 4 (1 representing no convenience, escalating to 4, representing convenience) on four different levels:

(1)  Clarity of content: the items are written clearly and precisely, which facilitates their understanding by the students.
(2)  Clarity of vocabulary: vocabulary and terminology are age-appropriate for the students.
(3)  Grouping of the questions: the content of the item and the category in which it is located correspond, and they are presented in a meaningful order.
(4)  The relevance of the data provided: the items are relevant and provide the necessary information to give answers to research objectives.

All items ($N = 43$) valued by expert judgment ($N = 6$) in the four dimensions (144 judgments) had arithmetic means above 2.5. As ratings range from 1 to 4 and the Archimedean point is 2.5, the items achieved high recognition in the opinion of the experts in the four evaluated dimensions. However, the real objective was to calculate a coefficient to understand if the experts answered with homogeneity. For the above reason, we decided to calculate the coefficient of the ambiguity of each item, which is the interquartile course (P75—P25) of each item [34]. For the interpretation, we used the criteria indicated by Barbero et al. [34]:

(1)  If the coefficient is between 0 and 1, the item remains as it is.
(2)  If the coefficient is between 1 and 2, the item should be revised for a possible restatement.
(3)  If the coefficient is > 2, the item must be rejected.

In our case, all 144 judgments' scores were below or equal to the interquartile range value of 1.5. Although few had scores between 1 and 1.5, they should be thoroughly reviewed for reformulation in the interests of a future study. The second reason for the validation by experts was to determine the degree of agreement reached in their assessments. To do this, we calculated the intraclass correlation coefficient (ICC) associated with the several bivariate variance analyses of repeated measures. Each valued item constitutes a level or source of variation of the factor items (within people) broken down into three more sources (between items, residual, and total) and each of the experts represents a level of the factor judges (between people). The results are provided in Table 3.

**Table 3.** Values of the intraclass correlation coefficients, F, and associated significances in each valued dimension.

| Dimension | Intraclass Correlation | F Value | Statistical Significance |
|---|---|---|---|
| Content | 0.57 | 2.33 | 0.000 *** |
| Wording | 0.55 | 1.96 | 0.010 ** |
| Understanding | 0.67 | 2.73 | 0.000 *** |
| Relevance | 0.68 | 2.16 | 0.001 *** |

Note: * $p < 0.05$, ** $p < 0.01$, and *** $p < 0.001$.

Regarding the obtained results, the four scale values of the interclass correlation coefficients (ICC) were 0.57, 0.55, 0.67 and 0.68 (single measures) of content, writing, comprehension, and relevance dimensions, respectively. For the four dimensions, according to Landis and Koch [35], we obtained moderate agreement values, and the F values in the implemented analysis of variance were associated with probabilities $p < 0.001$ and $p < 0.01$, which were considered statistically significant. However, the agreement is moderate and this could not be due to chance.

*4.3. Results of Concurrent Criterion Validity*

In the calculation of concurrent criterion validity, we correlated each item with the total test, except for the item itself, which is known as corrected item-total correlation and the multiple quadratic relationships or coefficient of multiple determination. Since we only had one application and no other instrument for comparison, this was the most appropriate option. The results are presented in Table 4.

**Table 4.** Values of item-total corrected correlations and multiple quadratic correlations obtained for each item on the validated scale.

| Attitude Toward Refugee Minors | Corrected Item-Total Correlation | $R^2$ |
|---|---|---|
| Refugee minors can spread diseases to schools in my country. | 0.348 | 0.285 |
| If I adopted a child, it would bother me if they were from another country. | 0.407 | 0.295 |
| It would be a problem if my son/daughter had a refugee child as a classmate. | 0.639 | 0.610 |
| Refugee children cause trouble during class. | 0.496 | 0.421 |
| It would be a problem if my son/daughter had a refugee child as a friend. | 0.611 | 0.574 |
| All people should be respected the same. | 0.465 | 0.336 |
| Refugee minors have the right to talk about their religion. | 0.629 | 0.641 |
| Refugee children have the right to celebrate their religious holidays in my country. | 0.679 | 0.662 |
| The refugee minors, in the future, will want to impose their religion on us. | 0.485 | 0.362 |
| We have to respect people from other countries. | 0.573 | 0.458 |
| The children of the refugees are inferior to the children of the Spaniards. | 0.437 | 0.328 |
| Refugee children have the right to use their native language. | 0.601 | 0.459 |
| Our country should expel all refugee children. | 0.571 | 0.465 |
| Thanks to refugee children, we can learn about other cultures and languages. | 0.713 | 0.786 |
| Thanks to refugee children, we can learn new things. | 0.735 | 0.826 |
| All refugee children who come to my country are poor. | 0.220 | 0.149 |
| Thanks to refugee children, we can learn new customs. | 0.692 | 0.708 |
| We must behave toward refugee children in the same way we do for Spaniard children. | 0.588 | 0.450 |

**Table 4.** *Cont.*

| | Corrected Item-Total Correlation | $R^2$ |
|---|---|---|
| Refugee children will cause an increase on crime rate in my country. | 0.580 | 0.554 |
| Refugee children arrive to my country only for economic reasons. | 0.346 | 0.247 |
| Refugee children have the right to attend school classes along with Spaniard children. | 0.700 | 0.615 |
| Public services should focus more on welcoming refugee minors. | 0.532 | 0.449 |
| Schools should have educational programs to meet the needs of refugee minors. | 0.697 | 0.651 |
| All children have the right to education. | 0.596 | 0.601 |
| **Feelings Toward Refugee Minors** | **Corrected Item-Total Correlation** | $R^2$ |
| I feel admiration toward refugee minors. | 0.487 | 0.412 |
| I feel hate toward refugee minors. | 0.215 | 0.296 |
| I feel compassion toward refugee minors. | 0.420 | 0.329 |
| I feel sorry toward refugee minors. | 0.232 | 0.262 |
| I fear refugee minors. | 0.319 | 0.135 |
| I feel solidarity with refugee minors. | 0.411 | 0.284 |
| I feel indifferent toward refugee minors. | 0.366 | 0.225 |
| I am curious about refugee minors. | 0.301 | 0.319 |
| I feel concern toward refugee minors. | 0.520 | 0.496 |
| I dislike refugee minors. | 0.151 | 0.395 |

Except for five items, the rest had correlation values higher than 0.35, indicating the direction and magnitude of the relationship of each item with the total test direct, which, in many cases, was moderately important.

The construction and validation of the attitudes scale was based on previous studies [20,21,35–43].

## 5. Discussion, Conclusions, and Suggestions

Our fundamental objective, which was achieved, was to design, elaborate, and validate a stable and well-structured scale evaluating attitude toward refugee children (AQReC) with improved validity and reliability regarding the information collected on this subject, as there is a lack of such tools.

Results from the Classical Test Theory (CTT) and the Generalizability Theory (GT) indicated that AQReC scale accumulates enough empirical evidence, achieving the research objectives. Furthermore, AQReC is valid and reliable. AQReC subscales had good reliability values, such as sufficiently high internal consistency, which were lower in the case of the feelings subscale. According to the values obtained by its application in the sample, we conclude that AQReC is an instrument in which the items have more in common than the differences between them. For these reasons, it is considered a stable and accurate instrument.

Concerning the degree of convenience of the items measured on the scale and the validity of the instrument, different typologies from the Classical Theory of Test (CTT) and the GT were considered, considering the four following dimensions: clarity of content, clarity of vocabulary, grouping of the questions and the relevance of the data provided. In all cases, the arithmetic means (all above the Archimedean point of 2.5), as well as the ambiguity indexes or calculated interquartile paths (all below the criterion value of 1.5), support the appropriateness and homogeneity of the instrument. The ICCs obtained from the facet analysis of the GT confirmed that the qualifications agreement is not random. Secondly, concerning the criteria validity, most of the items were measured individually. In this sense, most had total-item correlation values above 0.35. However, a few items had corrected total-item correlation coefficients (in the two subscales) that yield *r* values below 0.35 and should be reconstructed for future research since the correlations above 0.35 are statistically significant beyond the 1% level [20,21].

As mentioned before, after a literature review on an international level, we did not find any attitude scale questionnaire that measures host countries´ citizens´ attitudes toward refugee children; thus, we conclude that our study is the first one that focuses on this topic. Through this study, we provide new information and advances in human knowledge to the scientific community and the outside world.

As education is considered a priority for the 2030 Agenda for Sustainable Development, the AQReC fills the gap in the theoretical framework and responds to the question as to whether the millions of school-aged refugee children worldwide are welcome not only in host countries but also in the pubic educational centers. By applying the questionnaire, researchers can identify whether refugees fully enjoy their fundamental rights according to the 1951 Refugee Convention and the United Nations Convention on the Rights of the Child (1989).

The current study is the foundation for future research as researchers can apply the AQReC to a wider sample and in different countries. It would be interesting to apply such research among the teaching staff of the educational centers that will receive refugee children to their classrooms to discover the attitudes of the professional personnel. The AQReC became the inspiration for our scientific team to construct a second attitude scale questionnaire, developed to measure Spanish and Greek pupils' attitudes toward refugee children [8,14].

Regarding the methodology, it would be important to use the AQReC scale to determine the validity through some multivariate test of interdependence and then the dependence of dimension reduction. In this case, it would be interesting to initially calculate an exploratory factor analysis that would provide us a perspective on the set of aspects, factors, or components that individually constitute each of the two subscales that we proposed. Upon obtaining this information, it would be advisable to develop a confirmatory factorial analysis using a path analysis within structural equation modeling (SEM). The purpose of this technique is to corroborate the initial factorial structure of the exploratory factor analysis and determine the level of adjustment that each of the endogenous empirical variables contemplated (the 34 items) maintained (correlate and covariate) with the possible exogenous latent variables (dimensions, factors, or components inferred in the exploratory factor structure).

## 6. Limitations

Although we achieved our objectives, we should mention some unavoidable limitations. First, the literature review is considered very important for any study, as it provides the researchers with the opportunity to identify the scope of previous works. Thus, the lack of previous studies on host countries' citizens' attitudes toward refugees, as well as the absence of information collection methods, are considered limitations, as we did not have the chance to provide a duly justified theoretical framework.

**Author Contributions:** Conceptualization, G.A. and C.R.-S.; methodology, G.A., E.M.A.-R., and C.R.-S.; software, C.R.-S.; validation, C.R.-S.; formal analysis, C.R.-S.; investigation, G.A.; resources, E.M.A.-R. and G.A.; writing—original draft preparation, G.A.; writing—review and editing, G.A., C.R.-S., and E.M.A.-R.; visualization, G.A. ; supervision, E.M.A.-R. and C.R.-S.

**Funding:** This research received no external funding.

**Acknowledgments:** Special thanks to all the people from Greece, Spain, and Puerto Rico who agreed to participate in our research. We thank our colleagues from the University of Granada who provided insight and expertise that greatly assisted with our research. We thank the professors from the University of La Laguna, the University of Deusto, the University of Barcelona, the University Jaume I, the University of Jaen (Spain), and the University of Dimokritos (Greece) for their support that significantly improved our Attitude Questionnaire for Refugee Children. We would like to express our gratitude to Themelis Tourvas from the University of Essex, Tamer Abou El Alla from Panteion University, and Pierette Bartolomei Torres from the University of Granada.

**Conflicts of Interest:** The authors declare no conflict of interest.

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
