# Peer review of "Design and Validation of a Scale Measuring Attitudes toward Refugee Children"

_sustainability, doi:10.3390/su11102797_

Round 1

Reviewer 1 Report

Thank you for the work that you did. The topic discussed is very interesting and relevant to the field.

You will find below my comments that I hope will help you while revising the manuscript:

Introduction:

24-28: If the authors wish to provide an introduction to the European crisis they will need to add more details and include updated figures and references sources such as UNHCR and IOM. They do not have to provide a lot of information as this is not the main focus of the manuscript but rather more detailed synopsis of the crisis

30-33: this transition from the European crisis to the global situation could possibly confuse readers. For instance, has Puerto Rico expressed a desire to host refugees displaced by the Syrian war? As the authors decide to introduce the European migration crisis in the first paragraph of the manuscript, I would suggest to provide some information about the European countries that have engaged in the resettlement of refugees and present updated numbers. In this way they will be able to provide a stronger argument that as the number of refugee children being resettled in different countries increases, the need to overcome real challenges to a successful and sustainable integration and ensure that equal access to education is realized.  

34-46: In this paragraph, authors make an effort to explain the rational of their study. In my opinion, this paragraph would benefit from a discussion of the anti-immigrant rhetoric currently thriving in Europe, by providing significant examples and references (the examples of the island of Lesvos is not representative of the whole situation) and discuss why taking into account public opinion and social attitudes is so crucial when trying to integrate refugee children into the educational system. The authors can argue that integration is a two-way inclusion process that requires both the effort of the refugee population as well as the acceptance of the local population. I would also suggest to avoid any reference to the American system (North and South America have different policies in general) as there is no previous or later discussion on this. 

44-46: In my opinion this small paragraph would benefit if the authors could explain why this questionnaire will be so useful. Who could use it and why?

Literature review Summary:

48-49: “The literature review leads us to say that different authors [5, 6, 7, 8, 9, 10, 11] have identified a series of factors related to individual and group discrepancies between individuals and families.” In my opinion it is not very clear to the reader what kind of discrepancies the authors talk about. I would suggest to provide some details and perhaps some examples. Also it is not clear if they talk about refugee individuals and families or non-refugee individuals and families.

50-51: I would suggest to the authors to talk a bit about the “experience of the resettlement”, particularly as they mention that “these experiences must be taken into account”. This is a good place to also talk more about the two-way inclusion process of the integration of refugees. 

59-63: I would suggest to the authors to avoid presenting recommendations to the professionals while presenting the existing literature. In my opinion this kind of recommendations can be more useful in the discussion section. 

64: The authors mentioned “the research”. Which research? Do they talk about their research or do they talk about the research they reference at the end of the paragraph? It would be important for the reader to have a clear understanding of this.

78-79: “Their performance is lower than the performance of the rest of the students.” In my opinion, it would be better to avoid such generalizations. The authors can say” their performance tends to be lower…..”

48-88: I would suggest to the authors to revise these paragraphs. In my opinion, the arguments presented, though important, lack a more critical discussion. These paragraphs would benefit from a discussion of these arguments, while linking them to the aims of their study. 

91-95: the authors explain how critical it is for the livelihood of the refugee communities to have access to training programs. However, it is not clear how this argument supports their next argument about the need of examining the attitude of the host community (96-97). I would suggest to the authors to make this connection very clear and perhaps talk more about the challenges that the refugee communities face when they have to face a hostile local community. 

123-131: In my opinion the authors should discuss more the gap in literature and why and how their study can help addressing this lacunae. The reference of the study of Beverluis and the comparison with the current study and then the list of some of the topics that some existing tools examine cannot build a strong argument around the need to conduct this study. 

127: If Puerto Rico is one of the countries of interest for the authors they need to provide a background about the situation there. They need to clarify if Puerto Rico will resettle Syrian refugees (as the authors have presented this refugee crisis only) or they talk about a different situation there. In my opinion, authors should also explain why they have chosen the specific countries they focused on. 

133-134: I suggest the authors use either “children” or “minor” in a consistent way throughout the manuscript. I would also urge them to use the work “Children” over “minors”, but of course this is their decision to make. 

136-137: In my opinion, authors should also explain how a scale about the attitudes of the host community against refugee children can inform, guide and shape interventions tailored to the unique needs of children: “intervention plans, exclusively related to the children and their skills”

137-138: “currently used” by whom? 

Materials and methods:

153: “Attitudes Questionnaire towards Refugee Children (AQReC)”. Is this a questionnaire that the authors created for the needs of this study? It is not very clear to the reader in this paragraph.

3.2: In my opinion, the authors should provide more information about the sampling methodology they employed. What process did they follow to reach out to these participants? How did they decide this sample size? Is this sample size enough to yield representative results of the three nationalities, as well as of both sexes and all age groups? More information is needed so the reader can evaluate the merit of the sampling methodology employed. 

3.3 In my opinion, more information is required about the way the authors created this instrument. They need to present the procedure they followed in creating the specific items (eg what kind of literature they used, what kind of themes they decided to present, etc.). 

3.4 I would suggest to the authors to provide more data about the pilot phase. Did they pilot-test this instrument with the participants in each country as well? How can they ensure that this instrument is culturally appropriate for each of the three countries?

Results

4.1 I would suggest to the authors to also discuss the limitations of the methodology they used. A high a coefficient isn’t always the mark of a reliable set of items. A large number of items (as in the case of the attitude dimension) can also increase the coefficient. I would also add that a good scale can also be defined based on the theoretical background it is built on. This is extremely important for the readers to understand better how the authors created the scale. 

Discussion, conclusions and limitations

271: I would advise the authors to describe these very few existing tools and explain how their tool will fill in the gap in literature. 

297-307: the manuscript would benefit from a thorough discussion of the limitations of this study on top of the future work that can be conducted. It would also be interesting to the readers if the authors could discuss whether this tool can be used in different contexts and across different populations and what the limitations and the challenges could be. 

Author Response

Dear Reviewers,

Thank you for your letter and constructive comments concerning our manuscript entitled “Design and validation of an attitude scale towards refugee children”. We have studied your comments carefully and made major correction which we hope meet with your approval. I would like to inform you that we have changed almost totally the article, regarding your comments and for this reason, as you could see, there are no comments on the manuscript, as we worked on it in an holistic way.

First of all, we improved the introduction, the method explanation and the conclusions.

Also, we included the UN Agencies in our references (UNHCR, UNICEF, IOM), as well as information and data about migration and refugee issues, including the case of Puerto Rico and we clearly stated why we applied the research in these three specific countries (Greece, Spain and Puerto Rico). Moreover, we provided information about the resettlement scheme, the inclusion, which is a two way process and the gap in the literature review.

We revised all the paragraphs that you have suggested to. We should also say that it was a mistake from our part, as we mentioned only the Syrians as refugee population. Our study is about all the refugees who were forced to leave their countries. We included also information about the method, the type of the sample, the elaboration and evaluation process of the instrument and we provide information about the pilot phase.

Last but not least, we revised the discussion with the conclusions and we included the limitations of the work as well as the future lines of investigation.

We hope the revised manuscript will better suit the Sustainability  but also we are happy to consider further revisions, and we thank you for your continued interest in our research.

I am at your disposal.

Yours sincerely,

Georgia Angelidou

Reviewer 2 Report

The idea of creating a scale of this type is very interesting, however I believe that the limits proposed in the study (the proposed analyzes) should already be presented in a scientific article rather than being proposed as limits of work. Therefore, I believe that authors should supplement the article with these analyzes before it can be published.

Author Response

(The authors gave the same response as above.)

Round 2

Reviewer 1 Report

The authors have done a great job revising the introduction and the methodology. As a final advice, I'd suggest to the authors to thoroughly screen the manuscript for grammar and syntax mistakes. In my opinion, the manuscript can be considered for publication.   

Author Response

Thank you for your suggestions.

Regarding the conclusions and limitations we made the changes, as you suggested. Also, As the reviewers have suggested that our manuscript should undergo English editing, our scientific team has decided to send the paper to the MDPI author services.

Reviewer 2 Report

Dear authors,, the work has been revised but it seems to me that for what concerns the discussion and the limits the observations made previously were not taken into consideration. I would therefore ask you to review it in this regard

Best regards

Author Response

(The authors gave the same response as above.)
